# Drug-Induced Enterocolitis Syndrome in Children

**DOI:** 10.3390/ijms24097880

**Published:** 2023-04-26

**Authors:** Paola Di Filippo, Annamaria Venanzi, Francesca Ciarelli, Beatrice Panetti, Sabrina Di Pillo, Francesco Chiarelli, Marina Attanasi

**Affiliations:** Department of Pediatrics, University of Chieti, 66100 Chieti, Italy

**Keywords:** drug-induced enterocolitis syndrome, food-induced enterocolitis syndrome, non-IgE-mediated allergy, children, FPIES, DIES

## Abstract

Drug-Induced Enterocolitis Syndrome (DIES) is a drug-induced hypersensitivity reaction non-IgE mediated involving the gastrointestinal system that occurs 2 to 4 h after drug administration. Antibiotics, specifically amoxicillin or amoxicillin/clavulanate, represent the most frequent drugs involved. Symptoms include nausea, vomiting, abdominal pain, diarrhea, pallor, lethargy, and dehydration, which can be severe and result in hypovolemic shock. The main laboratory finding is neutrophilic leukocytosis. To the best of our knowledge, 12 cases of DIES (9 children-onset and 3 adult-onset cases) were described in the literature. DIES is a rare clinically well-described allergic disease; however, the pathogenetic mechanism is still unclear. It requires to be recognized early and correctly treated by physicians.

## 1. Introduction

In 2014, an Italian study group reported the case of a 6-year-old girl with a history of vomiting 1–2 h after amoxicillin (AMX) syrup intake and a history of urticaria at the age of 1 year after cefixime intake. Since the clinical manifestations and blood tests were similar to Food Protein-Induced Enterocolitis Syndrome (FPIES), the term Drug-Induced Enterocolitis Syndrome (DIES) was coined [1]. DIES is a rare non-IgE-mediated allergy syndrome with clinical similarities to FPIES. The symptoms consist of repeated delayed vomiting, diarrhea, and hypotension, which may progress to dehydration and hypovolemic shock [2].

In 2017, two more cases due to AMX were described in Spain: a child [2] and an adult [3].

In 2019, a Dutch study group published a fourth case of DIES after AMX in a 4-year-old boy, proposing specific diagnostic criteria based on FPIES ones [4].

To date, 12 clinical cases were described in the literature: 9 children-onset and 3 adults-onset DIES. The responsible drugs were AMX or amoxicillin/clavulanate (AMX/CLV) in 10 cases [1,2,3,4,5,6,7,8], pantoprazole in an adult case [3] and paracetamol in an infant [9]. 

Mean age at the symptoms onset in children with DIES (6.3 years of age) is higher compared to that of children with FPIES and ranges from 10 months to 14 years of age [1,2,4,5,6,7,9].

Differently from FPIES, DIES often occurs in individuals who have previously tolerated the causative drug [7]. Moreover, because of the few identified cases, time of recovery from the disease is well established for FPIES but still unknown for DIES [7]. 

The incidence of DIES is still not defined. In an Italian Children’s Hospital Allergy Unit, it was estimated to be equal to 0.4% (3 of 714 total cases of children referred for suspected hypersensitivity reaction from 2014 to 2019) [7]. A significant association between drug use and bowel function was demonstrated; drug-associated gastrointestinal events are the most commonly reported adverse drug effects [10]. Nausea, vomiting, and diarrhea are considered common side effects of AMX [11]. In this case, it is plausible that AMX was no longer administered, and another antibiotic was preferred, without performing a drug provocation test (DPT) and establishing a precise diagnosis. Therefore, the incidence of DIES is still underestimated and clinical awareness on DIES should be improved.

A cross-reactivity among drugs could be postulated. In the literature, only a few case reports about cross-reactivity among drugs were published. Infante et al. [2] discovered tolerance with penicillin V in a 3-year-old boy with DIES after AMX intake, suggesting that the side chain was responsible for the reaction. Worcel J et al. [6] reported tolerance with both penicillin V and cefpodoxime in a 10-year-old girl. Freundt Serpa F et al. [8] confirmed tolerance with penicillin G and V in an 18-year-old man. It would be interesting to investigate in studies including patients with confirmed DIES the cross-reactivity between antibiotics, in particular between penicillins and cephalosporins, which differ by side chains.

The pathophysiological mechanism is unknown, but the drug’s reactive metabolites or drug–protein complexes could directly or indirectly affect the gastrointestinal epithelium by an immunological reaction [7]. The main laboratory finding is neutrophilia, which can be related to cytokine and chemokine secretion due to the inflammation. Thrombocytosis and an increase in methemoglobin levels were also reported [1].

DIES is highly probable if persistent vomiting occurs 1 to 4 h after drug ingestion in the absence of classic IgE-mediated skin, and respiratory symptoms and at least three minor criteria as described for FPIES (i.e., pallor, lethargy, need for emergency department, diarrhea, hypotension, and need for intravenous fluid support) are present [4]. Currently, the diagnosis of DIES is based on clinical diagnostic criteria and is occasionally confirmed by DPT in daily clinical practice. Although not required for the diagnosis, a DTP is strongly recommended if a single episode occurred in order to properly confirm or exclude the diagnosis and to reduce the risk of overdiagnosis.

Similar to FPIES, intravenous fluid and antiemetics (ondansetron) for the treatment of vomiting appear to be effective, while adrenaline appears useful in some cases for hypotension [4].

## 2. General Mechanisms of Gut Protection and Injury

DIES includes morphological and functional alterations of the small and large intestine as a consequence of the exposure to pharmacological active compounds [10].

The enterocolitis results from an imbalanced interaction between the drug molecules and the tolerance of the intestine towards harmful insults [12]. The hypothesis is that alterations of the intestinal barrier facilitate the entry of drug antigens into the lamina propria with subsequent activation of the immune system and initiation and/or maintenance of inflammatory responses of the intestine [12].

Several factors play a key role in the pathologic gut response to drugs: age, gender, drug dose, exposure time, drug–drug and drug–food interactions [13].

However, the intestinal barrier has attracted increasing attention in recent years. The intestinal barrier is a complex system acting as a major line of defense, working in concert with immune cells to the elimination of pathogens and noxious substances [14]. Due to its ability to harbor up to 70% of lymphocyte population, the intestinal barrier is de facto the largest immunological organ in the human body [15]. Together with its immune system, it is constantly challenged with antigens (self-antigens, food, microbes, drugs, toxins), which continually shape host immunity [12]. The intestinal mucosa acts as a filter, limiting the number of luminal antigens that come into contact with the underlying immune system. Minimal alterations of components of the gut barrier can enormously influence the individual responses to substances entering the intestine, such as drugs [16]. However, ingestion of antigen mostly induces host tolerance through a subset of intestinal dendritic cells (DCs) that are able to promote the development of regulatory T cells [17].

The lamina propria of the gut villous epithelium contains a large population of DCs with two predominant groups (CD103+ and CX3CR1−DCs) which perform numerous activities including IgA production, the promotion of tumor necrosis factor-α (TNF-α), and development of TH17 T cells [18,19,20,21] acting synergistically in maintaining the homeostasis of the barrier.

Physiologically, the luminal surface of the gut barrier is coated with a hydrated gel produced by goblet cells (GCs) which prevents large particles and bacteria from coming into direct contact with the epithelium [14]. Van der Sluis et al. [22] conducted a study with knockout mice lacking mucin-2 (Muc2) protein, the major component of intestinal mucin, showing that Muc2-deficient mice spontaneously develop colitis [22]. Furthermore, using a minimally disruptive in vivo imaging approach, McDole et al. [23] showed that small intestine goblet cells GCs deliver soluble antigens from the gut lumen to underlying CD103+−DCs, implying a GC key role in intestinal immune homeostasis. Furthermore, several studies showed that the mucous layer undergoes a progressive maturation during childhood and that disturbances during this process could predispose to various intestinal diseases [24,25,26]. It could be hypothesized that children who develop DIES may present alterations in the development of the mucous layer, which predisposes to an abnormal response to drugs. In this view, glucocorticoid hormones seem to be able to partially induce the glycosylation of mucosal layer proteins in rat models [27,28] and increasing evidence has shown that the microbiota plays a fundamental role in the maturation of the mucosal coat [29,30,31,32].

The intestinal microbiota is a dynamic system comprising trillions of bacteria, viruses, archaea, eukaryotes, protozoa and their collective genome (called “microbiome”), playing a pivotal role in maintaining the integrity of the intestinal barrier [33,34]. Due to the release of substances (such as metabolites, peptides, and hormones) activating the immune system, the microbiota influence host immunity and metabolism, arising as a “new virtual metabolic organ” [35,36,37].

Recent studies have hypothesized that individual perturbations of the microbiota composition may be implicated in a different susceptibility to the onset and extent of intestinal damage induced by drugs [38], favoring the onset of DIES.

These hypotheses are further supported by the evidence that early childhood dysbiosis is linked with FPIES. A cohort study by Su et al. [39] showed that fecal samples from children with FPIES contained significantly lower levels of Bifidobacterium adolescentis and more pathogens, such as *Bacteroides* spp. and *Holdemania* spp., compared to those of healthy children. In addition, short-chain fatty acid (SCFA) levels were significantly lower in the stool of children with FPIES. SCFAs are produced by bacterial fermentation of indigestible carbohydrates and seem to play a key role in preserving the functionality of the intestinal barrier and counteracting the onset of inflammatory reactions due to the regulation of transcription of TJs proteins, especially claudin-1 [40,41]. SCFA modulates also host immune responses, regulating the number and functions of regulatory T cells [42,43,44]. These data suggest that dysbiosis and lower levels of SCFAs may play a role in the pathogenesis of FPIES and DIES. It would be interesting to investigate the influence of the gastrointestinal microbiota in the mucosal damage of drugs in an attempt to understand whether the restoration of eubiosis could play a role in modifying the natural history of DIES.

Understanding the molecular mechanisms of gut protection and injury could be crucial in future, particularly from a prevention and therapeutic perspective.

## 3. Possible Pathogenetic Mechanisms of DIES and Similarities with FPIES

The exact immunological mechanisms underlying DIES are still poorly characterized. To date, no studies investigating the molecular structure of drug allergens responsible for the enterocolitis have been reported in the literature. The release of toxic or immunogenic metabolites after first-pass hepatic and intestinal processing of the drug could cause the drug-induced intestinal damage [11].

Generally, small chemicals, such as drugs, tend to bind protein receptors through non-covalent bonds, such as hydrogen bonds or electrostatic interactions [45,46]. In some cases, the chemical molecules can also interact with molecules other than the target receptor, representing the so-called “off-target” activity of the drug [46]. In particular, the analysis of drug-induced immune reactions showed that off-target bindings can stimulate some types of human receptors, such as highly polymorphic human leukocyte antigens (HLA) and T-cell receptors and thus be responsible for the appearance of symptoms of delayed hypersensitivity reactions [46,47,48,49,50,51,52,53]. These hypersensitivity reactions are closely related to the binding of the drug to protein or peptide molecules and to the formation of new antigenic determinants (hapten–protein or hapten–peptide complexes) [52,54]. Haptens can also activate the innate immunity and dendritic cells, inducing a new immune response to a drug [55,56].

Infante et al. [2] speculated that drug-derived haptens could be responsible for the immunologic reaction in DIES. For many years, allergic sensitivity to penicillins was thought to probably occur due to the beta-lactam ring. However, the tolerance towards phenoxymethylpenicillin in some reported cases [2,6,8] suggested that the side chain of AMX and not the b-lactam ring side is responsible for allergy to penicillins [1]. However, studies evaluating other type of drug hypersensitivity highlighted that the immune reactivity could be triggered by the nature of carrier molecules, suggesting other pathogenic mechanisms [57,58].

Considering the similar clinical presentation, it is conceivable that FPIES and DIES share similar pathogenetic mechanisms.

FPIES is considered a cell-mediated disorder; however, few studies investigated the role of immunity cells in FPIES with inconclusive results, questioning their role in this disorder [59,60,61,62,63,64,65,66,67]. As for FPIES, the neutrophilia in DIES might suggest a key role of neutrophils in the pathogenesis of these food/drug-induced enterocolitis. Caubet et al. [59] conducted a study characterizing the humoral and cellular immune responses to casein in 38 children with a previous diagnosis of FPIES caused by cow milk. Firstly, the authors confirmed the paucity of humoral responses to casein in patients with FPIES. They also observed increased serum level of Interleukin (IL)-8 after a positive oral food challenge test, confirming neutrophil involvement. Furthermore, the authors identified a potential involvement of peripheral antigen-specific T cell-derived pro-inflammatory cytokines (i.e., IFN-γ, TNF-α, and IL-9) known to regulate the intestinal barrier permeability. In particular, tumor necrosis factor (TNF), a proinflammatory cytokine produced and released by monocytes, macrophages, T cells, and mast cells [68], seems to be crucial in the regulation of pore and leak pathways of the epithelial barrier, regulating the intestinal permeability. Interestingly, the mast cell is the only cell that can store presynthesized TNF within granules and release it within minutes of antigen exposure during allergic responses [69,70]. Therefore, mast cells are the only readily available source of TNF in peripheral tissues [71]. TNF acts on tissue remodeling, increases vascular permeability and contributes to macrophage activation and recruitment of inflammatory cells due to the upregulation of adhesion molecules such as ICAM-1, VCAM-1 and P- and E selectins [72,73,74,75,76]. Moreover, Caubet et al. detected higher levels of IL-9, an interleukin closely associated with Th2 response, in children with FPIES compared to children with IgE-cow milk allergy, suggesting that IL-9 also might be involved in the pathogenesis through its influence on intestinal permeability [59]. The authors also hypothesized that TNF-α acts synergistically with IFN-γ by altering intestinal permeability, resulting in an increase in the amount of antigen uptake into the submucosa with further activation of antigen-specific T cells [59]. In this perspective, the lowest levels of TGF-β, which protects the intestinal barrier against the penetration of foreign antigens [62,77,78], could also be implicated in the pathogenesis of FPIES.

In addition, Chen et al. [79] demonstrated that mast cell-derived IL-9 played a pivotal role in intestinal anaphylaxis. These findings underline the possibility of mast cell involvement in FPIES, suggesting the disorder as a variant of intestinal anaphylaxis [79]. Taking these data into account a direct interaction of drugs with mast cells could also be hypothesized, which could degranulate in response to the drug binding, releasing large quantities of inflammatory mediators, such as TNF [45]. Indeed, in patients with FPIES, higher amounts of intestinal TNF-α and IFN-γ levels with lower expression of TGF-β were detected [59]. These data suggest that proinflammatory cytokines influencing intestinal permeability may play a key role in the pathogenesis of FPIES and that clinical similarities with DIES could be supported by a similar interleukins pattern [59].

Recently, Lozano-Ojalvo et al. [80] conducted a study using a metabolomics approach to identify novel pathways associated with FPIES reactions. Serum samples from 10 patients with FPIES were collected before, during, and after an oral food challenge and compared with those collected from 10 asymptomatic individuals. The authors reported that the levels of 34 metabolites, such as inosine and urates of the purine signaling pathway, were increased during oral food challenges performed on the patients with symptomatic FPIES [80].

Based on the results of the study, it can be hypothesized that the activation of the purinergic pathway during FPIES reactions could be a possible pathological mechanism linked with the inflammation and vomiting, triggering serotonin release from gastric and duodenal mucosa [80].

The growing knowledge of the pathogenesis of FPIES could be helpful in understanding DIES, too. Nevertheless, further studies are needed to determine the exact pathogenetic mechanisms involved in FPIES and DIES. Studies including the evaluation of cytokines in stool samples before and after challenge test could be useful.

We summarized the pathogenesis of DIES in Figure 1.

## 4. Clinical Manifestations of DIES

Eight children presented DIES after AMX or AMX/CLV administration, while only one child presented this syndrome after paracetamol intake. IgE-mediated reactions mostly occur within 1 h of drug administration [81]. Conversely, symptoms of DIES appear at least 1 h after drug administration and are probably caused by a T cells-driven mechanism [7].

Symptoms often started more than 2 h after last drug intake during DPT. Therefore, the onset often occurs after 2 h of surveillance without any clinical reactions and discharge [7].

Vomiting is the first manifestation of DIES and appears 1–4 h after the drug administration, without classic IgE-mediated skin and respiratory symptoms [4]. Children mostly presented vomiting after the drug administration, but some patients presented urticaria or angioedema. Therefore, IgE-mediated disease was first suspected. However, all children vomited after DPT.

Other frequently reported symptoms were lethargy (7/9 patients) [1,4,7,9] and pallor (7/9 patients) [1,5,7,9]. Less frequent symptoms were diarrhea (3/9 patients) which can occur 8–10 h after the drug intake [1,2], abdominal pain (4/9 patients) [2,5,7] and tachycardia (3/8 patients) [1,5,9].

In some cases of DIES, a skin involvement was observed during the first reaction [1,2,4,5,7]. In two infants with FPIES, a possible shift from an IgE to a non–IgE-mediated reaction was suggested [82]. However, the coexistence of both immediate and delayed mechanisms of reaction in the same patient is difficult to understand and skin manifestations during antibiotics in children is often due to a viral/bacterial infection [7].

Clinical and laboratory findings and therapeutic approach in the nine children reported in literature are described in Table 1.

## 5. Laboratory Findings in Children with DIES

The main laboratory findings are neutrophilic leukocytosis and an increase in methemoglobin [1,2,4,9]. Tryptase dosage was always normal [1,2,4,5]. These blood tests are not specific, but they can support the diagnosis, as in FPIES [83]. In FPIES, neutrophilia reached a peak at 6 h after the ingestion of the incriminated food; the increased levels of IL-8 and cortisol, released in response to stress, could be responsible for it [84]. In the described cases of DIES, blood tests were performed at different time points; therefore, although the peak at 6 h is also expected in DIES, it was not yet determined.

Methemoglobinemia was observed especially in patients with severe clinical presentation, and it is caused by the increased oxidation of iron in hemoglobin due to nitrites and nitrates released during inflammation [1,7]. Tryptase dosage was always normal in DIES [1,2,5,9].

To date, no validated biomarker of DIES is established. The only biomarker detected was described in a 18-year-old man: Freundt Serpa et al. [8] observed an increased eosinophil cationic protein in stool samples from 24 and 48 h after DPT with AMX/CLV. Studies in children are required to confirm these data.

During the diagnostic work-up, most children (8/9) underwent an allergy evaluation, which was always negative. Specifically, children underwent blood dosage drug specific IgE (3/8) [1,4,7], skin prick test (5/8) [2,4,7,9], and intradermal test (5/8) [2,5,7,9].

Atopy patch tests are commonly used to diagnose non-immediate T-cell-mediated drug hypersensitivity reactions [85]. Few studies evaluated the role of atopy patch test in diagnosis and follow-up of FPIES and DIES. In 2012, atopy patch testing was evaluated as predictive tool of tolerance development in a study including 25 subjects with FPIES at median age of 3.3 years [43]. Thirty-eight atopy patch test were performed before oral food challenge test. The authors determined that 16 of the 38 (42%) oral challenge tests were positive, and only two subjects had a positive atopy patch test. Among the 23 negative oral challenge tests, 2 subjects showed a positive atopy patch test. Therefore, atopy patch test had a sensitivity of 11.8%, a specificity of 85.7%, a positive predictive value equal to 40%, and negative predictive value equal to 54.5%. The authors stated that atopy patch test to common food allergens has poor utility in the follow-up prediction of outgrowing FPIES in children [43].

Interestingly, atopy patch test for AMX was evaluated in a child with DIES and resulted negative [7]. Nevertheless, positivity patch test for drugs ranges widely in literature, especially among children (0.9–89%) [35,36], and it was performed only in children.

To date, no specific recommendations on the utility of atopy patch tests have been established. It would be interesting to investigate the role of atopy patch test in the diagnostic work-up and follow-up of patients with DIES.

## 6. Diagnosis of DIES

Similar to FPIES, the diagnosis of DIES is based on clinical findings, and no biomarkers are currently available. The diagnosis is highly probable when protracted vomiting occurs 1 to 4 h after drug intake in absence of classic IgE-mediated skin and respiratory symptoms, in addition to at least three minor criteria described for FPIES (Figure 2) [86].

The latency between drug administration and symptoms onset needs to be considered in order to differentiate between DIES and type I hypersensitivity to AMX or AMX/CLV gastrointestinal symptoms. Indeed, the latency between drug intake and vomiting is >1.5–2 h in DIES, while in IgE-mediated drug hypersensitivity reactions gastrointestinal symptoms typically manifest within minutes or no later than 1 h. Moreover, vomiting is typically sustained and accompanied by a poor general condition in DIES, whereas vomiting secondary to IgE-mediated drug hypersensitivity reactions is typically associated with involvement of other systems, such as skin rash or swelling, due to histamine release [7].

Currently, the diagnosis of DIES primarily depends on clinical diagnostic criteria and is occasionally confirmed by a DPT. DPT appears to be the only test useful for validating or excluding DIES and evaluating tolerance to related and unrelated compounds.

After a DPT, observation for at least 3 to 4 h is recommended, as the latency between drug intake and symptoms onset ranges from 90 min to 4 h. However, the diagnosis of DIES is essentially a clinical diagnosis [7].

## 7. Management of DIES in Children

Management of DIES is mainly supportive. Similar to FPIES, treatment is expert opinion-based [86] and consists mostly of saline solution infusion, ondansetron and corticosteroids.

Almost all children required intravenous rehydration with saline solution, and only one child benefited from oral rehydration [6].

Ondansetron was administered to five children, with good clinical response and progressive improvement of vomiting [2,4,5,7]. Ondansetron is a serotonin 5-HT3 receptor antagonist used to treat emesis after chemotherapy, but also viral gastroenteritis. In retrospective cohorts, a reduction in the severity of vomiting in FPIES was demonstrated. It acts on central nervous system and peripheral nerves and its efficacy in FPIES suggests a neuroimmune mechanism in addition to the immune-mediated one. A dysregulation or limited compensation of the autonomic nervous system, depending on cytokine release and gastrointestinal losses, may be responsible for the exaggerated cardiovascular response and lethargy [87]. Immune cells are able to synthesize serotonin; in addition, mast and enterochromaffin cells have been identified as a source of gastrointestinal serotonin release [88]. Further studies are needed to determine the potential neuroendocrine mechanism of FPIES and consequently to understand DIES pathophysiology. Ondansetron is contraindicated in infants younger than 6 months of age and in children with cardiac disease, as it is associated with QT prolongation. A large retrospective study with 37,794 children observed seven cases of ventricular arrhythmia in children with a previous congenital cardiac conduction abnormality or other major cardiac diseases [89]. Nevertheless, routine electrocardiogram and electrolyte screening are not recommend [90].

Efficacy of corticosteroids is not proven, but in FPIES they are used in patients with severe symptoms because of the presumed cell-mediated inflammation [88].

In FPIES, adrenaline is not recommended, but it is often used as a pressure stabilizer if fluid resuscitation is unsuccessful [91]. Adrenaline has been shown to be effective in controlling hypotension in an adult patient with DIES [3]. In two children, adrenaline was administered as an initial misdiagnosis of anaphylaxis was established [4,5]. In the first case, the vomiting persisted despite administration of intramuscular adrenaline [4]. Therefore, ondansetron and hydrocortisone were intravenously administered, and the symptoms diminished. Approximately 90 min after the symptom onset, abdominal pain and vomiting started again; intramuscular adrenaline was administered a second time, with a following progressive improvement to a complete recovery 2.5 h after onset of symptoms [4]. In the second case, two injections of intramuscular adrenaline were administered in association to intravenously fluid. However, vomiting persisted and stopped after the ondansetron administration [5].

Table 1 shows the therapeutic tools used in the nine children with DIES described in the literature.

## 8. Conclusions

First described in 2014, DIES is a recently defined clinical entity. Given the scarcity of cases observed so far, only case series were reported in the literature. To date, AMX, AMX/CLV and paracetamol are the only known drugs causing DIES in children. It would be interesting to investigate whether other drugs can cause this syndrome in children.

Given their similarity, knowledge about DIES was mostly derived from FPIES. However, the incidence, underlying genetic factors, natural history and pathogenic mechanism of DIES are still unclear.

We suggest that incidence of DIES is underestimated in children. After a drug reaction, provocation testing is performed less frequently in childhood for diagnostic confirmation. In clinical practice, drug avoidance is preferred, and it is recommended eventually to perform provocation tests for alternative drugs. Furthermore, this syndrome is often unrecognized, and a misdiagnosis of drug allergy, drug adverse reactions or viral diseases is established. These factors lead to misdiagnosis and mistreatment, whereby these children are often labeled and treated as children with an IgE-mediated allergy.

The poor knowledge of the pathogenesis leads to diagnostic and therapeutic difficulties, but the clinical features of DIES are better defined in the literature. DIES should be suspected whenever a child presents gastrointestinal symptoms (especially repeated vomiting), lethargy and pallor within a few hours of drug administration. The diagnosis should be suspected especially in absence of skin and/or respiratory involvement and in case of neutrophilic leukocytosis, methemoglobinemia and negative allergy tests. Given the suspected cell-mediated pathogenesis, it would be interesting to investigate the diagnostic usefulness of atopy patch test in rigorous studies.

Further studies are needed to better understand pathophysiology of this syndrome leading to the development of useful diagnostic tools and to a better management of these children.

## Figures and Tables

**Figure 1 ijms-24-07880-f001:**
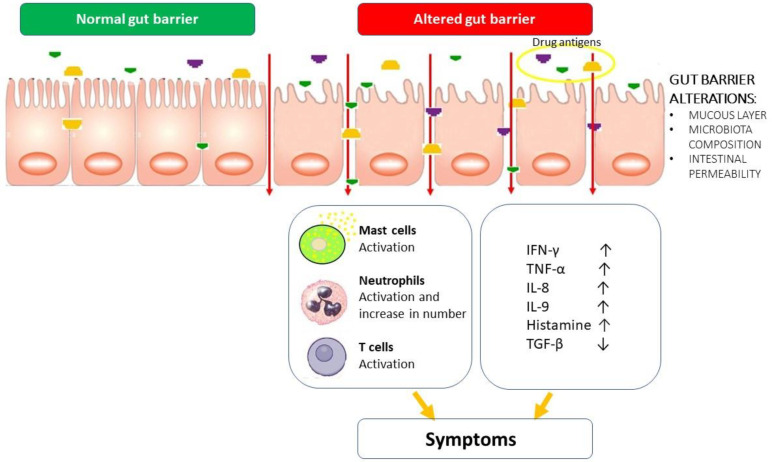
Pathogenic hypothesis of DIES. Similar to FPIES, alterations of the intestinal mucosa may favor the entry of drug antigens into the lamina propria with subsequent activation of immune cells, release of proinflammatory cytokines and histamine. The inflammatory response causes DIES symptom onset.

**Figure 2 ijms-24-07880-f002:**
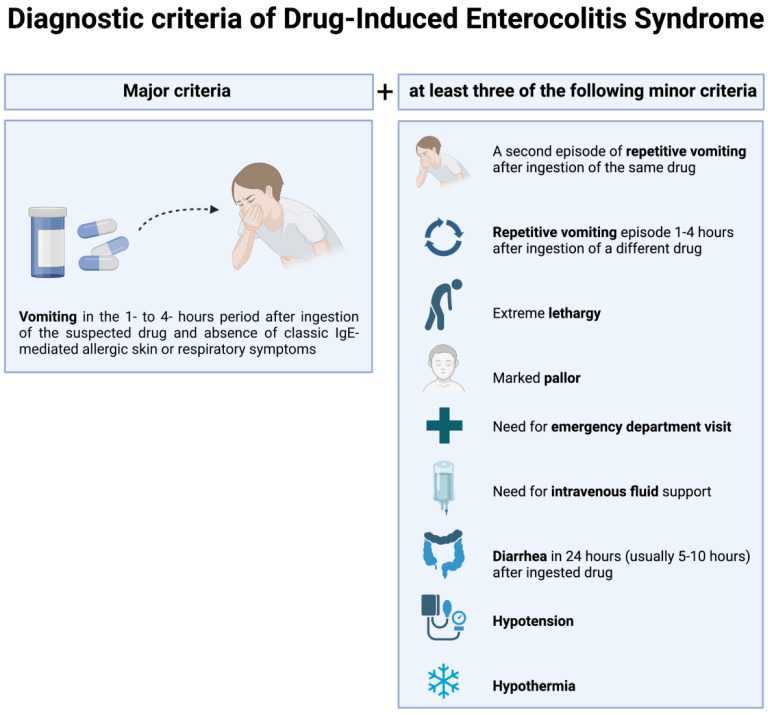
Diagnostic criteria of Drug-Induced Enterocolitis Syndrome. The diagnosis is highly probable when protracted vomiting occurs 1 to 4 h after drug intake in absence of classic IgE-mediated skin and respiratory symptoms, in addition to at least three minor criteria.

**Table 1 ijms-24-07880-t001:** Demographic data, clinical manifestations, and treatment in children with Drug-Induced Enterocolitis Syndrome described in the literature.

Study	Age (years)	Triggering Drug	Clinical Manifestations after Drug Intake	Clinical Manifestations after DPT	Time between Drug Administration and Symptoms	Laboratory Findings	Treatment
Novembre, 2014 [1]	6	AMX	Vomiting and a morbilliform rash the next day	Vomiting, diarrhea, pallor, lethargy	2 h	Leukocytosis with neutrophilia (WBC 20,480/mm^3^, N 82.2%) MethHb 1.1%	Saline solution infusion, IV hydrocortisone
Infante, 2017 [2]	3	AMX	Acute urticaria (2 y)	Vomiting, diarrhea, moderate abdominal pain	4 h	Leukocytosis with neutrophilia (WBC 17,800/mm^3^, N 73.6%)	Saline solution infusion, antiemetics, hydrocortisone
Van Thuijl, 2019 [4]	4	AMX	Repetitive vomiting and lethargy	Severe abdominal pain, vomiting, pallor, lethargy	1.5 h	Leukocytosis with neutrophilia (WBC 24,000/mm^3^; N 82.5%)	IM adrenalina, antihistaminic, IV hydrocortisone and ondansetron
Worcel, 2020 [6]	10	AMX	Repetitive vomiting, pallor, watery diarrhea 10 h later	Repetitive vomiting, pallor, lethargy, abdominal pain, watery diarrhea 8 h later	2 h	N/A	Desloratadine, prednisolone, oral rehydration
Mori, 2021 [7]	6	AMX/CLV		Vomiting, pallor, lethargy	2.5 h	Leukocytosis with neutrophilia (WBC 15,350/mm^3^, N 85.8%) MethHb 0.7%	Saline solution infusion
Mori, 2021 [7]	14	AMX/CLV	Persistent vomiting with streaks of blood and lethargy	Vomiting, pallor, lethargy, abdominal pain, dehydration	2.5 h	Leukocytosis with neutrophilia (WBC 10,730/mm^3^, N 86.3%) MethHb 0.7%	Saline solution infusion and IV ondansetron
Mori, 2021 [7]	9	AMX	Maculopapular exanthema	Vomiting, pallor, lethargy	3 h	N/A	Saline solution infusion and IV ondansetron
Pascal, 2022 [9]	0.8	PAR	Vomiting, asthenia, pallor, tachycardia	Vomiting, pallor, lethargy	4 h	Leukocytosis with neutrophilia (N 11,080/mm^3^) MethHb 1.3%	Saline solution infusion, steroid therapy
Eyraud, 2023 [5]	4	AMX	Erythematous rash and eyelid edema on the 7th day of treatment (2 y)	Vomiting, abdominal pain, intense pallor, and tachycardia	2 h	N/A	Saline solution infusion, IV ondansetron, IM adrenaline

AMX: amoxicillin; AMX/CLV: amoxicillin/clavulanate; PAR: paracetamol; WBC: white blood cells; N: neutrophils; MethHb: methemoglobin; IV: intravenous; IM: intramuscular; h: hour; N/A: not available.

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
