# Peer review of "Drug-Induced Enterocolitis Syndrome in Children"

_ijms, 2023, doi:10.3390/ijms24097880_

Round 1
Reviewer 1 Report
The paper is well-written, concise, comprehensive and very informative. All needed aspects of this condition are covered. The article is consistent within itself. The references are relevant and recent. The cited sources are referenced correctly. Appropriate and key studies are included. The flow of the paper is logical and the data is presented critically. The illustrative material is good and in enough quantity to present the key points of the paper.
However, there are some specific comments on weaknesses of the article and what could be improved:
Major points - none
Minor points:
1. Extend some of the titles to be more independent, for example - 3. Pathogenesis of DIES
2. The title of Table 1 should be above the table. Instead of "/" consider usign N/A (not available)
Author Response
The paper is well-written, concise, comprehensive and very informative. All needed aspects of this condition are covered. The article is consistent within itself. The references are relevant and recent. The cited sources are referenced correctly. Appropriate and key studies are included. The flow of the paper is logical and the data is presented critically. The illustrative material is good and in enough quantity to present the key points of the paper.
However, there are some specific comments on weaknesses of the article and what could be improved:
Major points - none
Minor points:
- Extend some of the titles to be more independent, for example - 3. Pathogenesis of DIES
ANSWER: We thank the reviewer for his/her comment. As suggested, we modified some of paragraph titles.
- The title of Table 1 should be above the table. Instead of "/" consider usign N/A (not available)
ANSWER: We thank the reviewer for his/her comment. We modified table 1 as suggested.
Reviewer 2 Report
The work of Fillipo et al. is of interest and brings a good contribution to the current state of knowledge in the field of pediatrics regarding the topic in question.
The title is concise and so is the abstract, although I would suggest that the statement "The key laboratory finding is neutrophilic leukocytosis" to be elaborated.
Line 67: "...and is occasionally confirmed by DPT" state citation as in when it was detected by DTP
Line 77 "The enterocolitis results from an imbalanced interaction between the effect of drug molecules and the tolerance of the intestine towards harmful insults." please cite and provide further details .
Line 90: "..individual responses to substances entering the intestine, such as drugs" provide citation
Overall a good work, I suggest only a minor revision taking in consideration the above stated.
Author Response
The work of Fillipo et al. is of interest and brings a good contribution to the current state of knowledge in the field of pediatrics regarding the topic in question.
The title is concise and so is the abstract, although I would suggest that the statement "The key laboratory finding is neutrophilic leukocytosis" to be elaborated.
Line 67: "...and is occasionally confirmed by DPT" state citation as in when it was detected by DTP
Line 77 "The enterocolitis results from an imbalanced interaction between the effect of drug molecules and the tolerance of the intestine towards harmful insults." please cite and provide further details .
Line 90: "..individual responses to substances entering the intestine, such as drugs" provide citation
Overall a good work, I suggest only a minor revision taking in consideration the above stated.
ANSWER: We thank the reviewer for his/her comment. We modified the manuscript following his/her suggestions.
Reviewer 3 Report
Paola Di Filippo, et al. reviewed the pathogenesis, clinical manifestations, laboratory findings, diagnosis and treatment of drug induced enterocolitis in children based on previously published case reports and literature, focusing mostly on the pathogenesis of DIES. The authors presented the current state of knowledge of DIES in a clear and well-organized manner. The publication will be useful to practicing physicians. My only suggestion to the authors is that to consider including few figures to summarize immunopathogenesis of DIES, and differences/similarities between DIES/FPIES since text is difficult to follow with multiple pathogenetic mechanisms involved.
Author Response
Paola Di Filippo, et al. reviewed the pathogenesis, clinical manifestations, laboratory findings, diagnosis and treatment of drug induced enterocolitis in children based on previously published case reports and literature, focusing mostly on the pathogenesis of DIES. The authors presented the current state of knowledge of DIES in a clear and well-organized manner. The publication will be useful to practicing physicians. My only suggestion to the authors is that to consider including few figures to summarize immunopathogenesis of DIES, and differences/similarities between DIES/FPIES since text is difficult to follow with multiple pathogenetic mechanisms involved.
ANSWER: We thank the reviewer for his/her comment. As suggested, we created figure 1 illustrating the pathogenic hypothesis of DIES.